# Extraction of Purple Prickly Pear (*Opuntia ficus-indica*) Mucilage by Microfiltration, Composition, and Physicochemical Characteristics

**DOI:** 10.3390/polym16233383

**Published:** 2024-11-30

**Authors:** María Carmen Fernández-Martínez, Cristian Jiménez-Martínez, Mónica Rosalía Jaime-Fonseca, Liliana Alamilla-Beltrán

**Affiliations:** 1Instituto Politécnico Nacional, Escuela Nacional de Ciencias Biológicas, Wilfrido Massieu s/n, U.P. Adolfo López Mateos, Gustavo A. Madero, Ciudad de México CP 07738, Mexico; 2Instituto Politécnico Nacional, Centro de Investigación en Ciencia Aplicada y Tecnología Avanzada, Legaria. Calz. Legaria 694, Col. Irrigación, Miguel Hidalgo, Ciudad de México CP 11500, Mexico

**Keywords:** prickly pear, natural polymers, *Opuntia ficus-indica*, mucilage, microfiltration, GC/MS

## Abstract

Mucilages are valuable to the food industry, but the solvents used to extract and concentrate them are detrimental to the environment. Therefore, environmentally friendly technologies that preserve the properties of biopolymers and reduce the use of solvents are being sought. In this work, the mucilage of *Opuntia ficus-indica* (mesocarp–endocarp) was extracted by two methods: In the first one, the pulp from the mesocarp–endocarp was extracted by ethanol precipitation and centrifugation cycles, then dried at room temperature. For the second, the pulp was processed in a three-step tangential microfiltration process: microfiltration (separation), diafiltration (purification), and concentration. The mucilages obtained differed significantly (*p* < 0.05) in color, betalains, total sugars, and proteins. The proportions of insoluble and soluble dietary fiber were similar. GC/MS analysis identified seven neutral sugars and a high content of uronic acids (31.3% in the microfiltered mucilage and 47.5% in the ethanol-precipitated mucilage). These show a low degree of esterification, which gives them a polar and hydrophilic character and the possibility of interacting with divalent ions through the carboxylic acid groups, which could form gels stabilized by an egg-box mechanism, with application as a thickening, stabilizing, gelling, or film-forming agent for foods with low sugar content.

## 1. Introduction

The cultivation of nopal (*Opuntia* spp.) is increasing its popularity around the world. Its unique characteristics give it resistance to drought, high temperatures, and poor soils [1]. Due to its high content of fiber, minerals, antioxidants, vitamin C, and betalains, the fruit of the nopal, known as tuna (chumbera, prickly pear, pita, higuera de pala, or palera), is an ideal source of nutraceuticals or functional ingredients [2,3,4]. 

Among the most abundant components of prickly pears are mucilages, which are heterogeneous polysaccharides produced naturally by plants [5]. They can be classified as non-conventional native gums, since they derive from non-traditional natural sources. Their primary advantages are their non-toxic property, flexibility, bioavailability, and biodegradability.

Opuntia mucilage has been shown to have several health benefits in a growing number of studies [6]. These include healing properties [7], anti-ulcer effects [8], and hypolipidemic activity [9], prebiotic [10], antioxidant [11], and antimicrobial [12] properties. In addition to their biological activities, Opuntia mucilages have several technological properties, making them ideal as thickening agents, emulsifiers, and stabilizers [9]. Due to these properties, they have been successfully applied in a variety of food products, including dairy beverages [13], breads [14], cookies [15], and mayonnaises [16].

Several mucilage extraction techniques have been developed or improved to increase its yield and improve its use, which should allow its application at the industrial level [17]. Extraction mainly involves conventional methods using organic solvents [4,5,6,18,19], but new techniques such as ultrasound-assisted extraction (UAE) and microwave-assisted extraction (MAE) have proven advantageous for mucilage isolation [20]. However, it has been reported that extraction of mucilages by methods involving extraction under acidic conditions or at high temperatures or microwave-assisted extraction can cause structural changes in the polysaccharides that affect their functionality. This may occur due to a decrease in molecular weight, degradation of the polymeric structure, and disruption of the formation of a polymeric network structure [18,21,22].

In this context, membrane separation processes, which are controlled by pressure gradients, have become an environmentally friendly alternative [23]. Membrane technology processes used in the food industry include microfiltration, ultrafiltration, nanofiltration, and reverse osmosis. These techniques offer several advantages, such as high separation and purification capacity, low energy consumption, easy modification of operating variables (pressure, temperature, or feed flow rate), and scalability [24,25,26]. Tangential separation membranes act as barriers that separate two phases and allow the selective transfer of matter between them. Depending on the stream containing the products of interest (permeate, retentate, or both), membrane treatments allow selective recovery of products by concentration, purification, or fractionation [21].

Mucilages are commonly found in various botanical parts of cacti, from flowers [12], cladodes [11,27,28,29,30,31,32,33], prickly pear peel [34,35,36,37,38,39,40,41], even the peeled fruit or mesocarp of *O. ficus-indica* [5,42,43]. In addition, Martins et al. (2023) [44] reported that about 70% of the total dietary fiber of prickly pear is found in the mesocarp of *Opuntia ficus-indica* fruits, and its consumption does not pose a risk related to heavy metals.

Therefore, we propose a novel process for the simultaneous extraction of mucilage from the mesocarp and endocarp of *Opuntia ficus-indica* using two environmentally and mucilage-friendly methods, at the natural pH of the fruit and without heat treatment for the extraction of the polysaccharides. The first method uses direct ethanol precipitation, and the second is based on microfiltration with tangential flow membranes followed by alcoholic precipitation of the mucilage. The extracted mucilages are subsequently subjected to physicochemical and structural characterization.

## 2. Materials and Methods

*Opuntia ficus-indica* cultivar San Martín was obtained from a commercial plantation in San Felipe Teotitlán, State of Mexico, Mexico (19°48′10″ N, 98°42′05″ W at 2453 m above sea level). The prickly pears were weighed, washed, and sanitized by immersion in sodium hypochlorite at 50 ppm for 5 min; then, the fruits were scalded with boiling water for 1 min and cooled in water at 0 °C; the prickly pears were then packed in polyethylene bags and kept frozen at–20 °C until use. The frozen prickly pears were peeled with a knife removing the epicarp (what we will call peel), avoiding as much as possible the removal of the mesocarp. The peel was retained on one side and the fruit with seeds and without peel on the other. Both fractions were subsequently weighed. The peeled and partially thawed fruits were passed through a pulper (185SC, Reeves Pulper Division Columbus, OH, EE.UU., thus separating the seeds and the pulp formed by the mesocarp–endocarp (MEP). Each fraction was weighed and recorded as a percentage of the fruit’s weight.

### 2.1. Characterization of the Mesocarp–Endocarp Pulp (MEP)

#### 2.1.1. Color Parameters

The CIE L*a*b* system was used to measure color. A colorimeter (ColorFlex D65/10°, HunterLab, Richmond, VA, USA) was used for liquid and powder samples. The instrument was calibrated with black and white standard plates. A colorimeter (CR-10, Konica Minolta, Osaka, Japan) was used for fresh prickly pears. From the color values, the chromaticity (C*), hue angle (H°), and total color change (ΔE) were calculated [45].
(1)$$C^{*}=\sqrt{a^{*2}+b^{*2}}$$
(2)$$H{}^{\circ}=arc\tan\frac{b^{*}}{a^{*}}$$
(3)$$∆E=\sqrt{{\left(∆L^{*}\right)}^{2}\pm {\left(∆a^{*}\right)}^{2}\pm {\left(∆b^{*}\right)}^{2}}$$

Color representation is a three-dimensional concept, and the data can be expressed in terms of the coordinates of a color space. Thus, the value of H obtained from Equation (2) is expressed in degrees ranging from 0° to 360° [46]. Depending on the value of ∆E, the color difference between the samples is classified as “not perceptible” (0–0.5), “slightly perceptible” (0.5–1.5), “perceptible” (1.5–3.0), “well visible” (3.0–6.0), and “very visible” (6.0–12.0) [47].

#### 2.1.2. Chemical Analysis

The proximate chemical analysis of the MEP and prickly pear peel was determined according to the official methods reported by the Association of Official Analytical Chemists [48]. Total acidity was determined in the MEP by potentiometric titration with 0.01 N NaOH (up to pH 8.2), and the results were expressed as grams of citric acid per gram of sample according to the following equivalence: 1 mL NaOH 0.1 N = 6.4 mg of citric acid. To measure pH, we followed the standard AOAC procedure (1990). Total sugars were analyzed using the phenol–sulfuric acid technique with a glucose standard [49]. The soluble solids content (°Bx) was determined with a benchtop Abbe refractometer with a scale 0–95 °Brix [48]. The total proximal betalain content was quantified by UV–vis spectroscopy [25] using a spectrophotometer (Genesys 10 S UV-VIS, Thermo Scientific, Beijing, China). The measurement of the maximum light absorption in the visible region at 476 nm was used to quantify betaxanthins and at 538 nm for betacyanins. All reagents used were analytical grade.

### 2.2. Methods for Separating Mucilage from Opuntia Ficus-Indica cv. San Martín

#### 2.2.1. Mucilage Separation by Direct Ethanol Precipitation (DEP Mucilage)

The separation of polysaccharides from MEP pulp was carried out by ethanol precipitation, modifying some of the methods already described in the literature [50,51,52]. To 1.9 kg of MEP, 96% ethanol was added at a 1:3 ratio (pulp/ethanol, *w*/*w*). Air was then bubbled by applying pressure for 5 min to promote the MEP–ethanol interaction. The resulting suspension was placed in 50 mL centrifuge tubes, left to rest in the refrigerator for 24 h, and centrifuged at 2500× *g* at 10 °C for 15 min (Allegra X-12R centrifuge, Beckman Coulter, Montreal, QC, Canada). The supernatant was decanted, and the precipitate resuspended in 96% ethanol (1:3 ratio); the suspending–resting–centrifuging cycle was repeated three times. The sediment was placed on a tray and dried in an oven at 30 °C for 48 h. The ethanol-precipitated dry extract (EPDE) was pulverized in a mill (BCG01, Bogner, Shanghai, China) until a particle size below 0.295 mm was obtained. The pulverized EPDE was sieved with a no. 50 mesh. The powder (DEP mucilage) obtained was stored at room temperature in an airtight jar for later analysis.

#### 2.2.2. Mucilage Separation by Tangential Flow Microfiltration (PP01 Mucilage)

The separation of mucilage by tangential flow microfiltration was carried out in three stages: mucilage separation, diafiltration to remove low molecular weight impurities, and concentration. 500× *g* of *O. ficus-indica* MEP was thawed under refrigeration at 6 °C. The pulp was then vacuum-filtered with a 0.351 mm mesh size (no. 40) to remove large particles before microfiltration. During separation, the tangential flow filtration unit was operated in dynamic mode; for this purpose, the sieved MEP was diluted 1:5 (*w*/*w*) in distilled water and was fed to the polysulfone hollow fiber module (CFP-1-E-4MA, GE Healthcare Bio-Sciences Corporation, Piscataway, NJ, USA with a nominal pore size of 0.1 µm and a surface area of 420 cm^2^) using a peristaltic pump (77602-10 Cole-Parmer, Vernon Hills, IL, USA) at 4.2 ± 0.03 L/h and 90 kPa. Two flow lines were derived from this filtration unit, one for permeate and one for retentate. The retentate line was of interest because it contains molecules over 0.1 µm, such as mucilages. The diafiltration stage was performed in discontinuous operation mode. The retentate was first diluted with distilled water 1:1 and then concentrated back to its initial volume. This process was repeated five times to remove low molecular weight impurities from the retentate [53], which were removed from the microfiltration system with the permeate. In the third stage, the retentate was concentrated by modifying the arrangement of the filtration system to a recirculation setup; the retentate was conducted into the feed tank to be pumped into the filtration module. Here, two streams further separated the permeate and the concentrate, which was sent back to the feed tank until its volume was reduced. The final concentrate was mixed with 96% ethanol 1:3 (*w*/*w*) to precipitate the mucilage, and centrifuged, dried, and ground as described for the EPDE. The final product of this process was the mucilage powder, separated by microfiltration using a 0.1 µm membrane (PP01 mucilage).

### 2.3. Characterization of Mucilage Powder

The physicochemical and structural characteristics of the mucilage powder obtained by direct precipitation with ethanol (DEP mucilage) or precipitation by tangential microfiltration (PP01 mucilage) were evaluated.

#### 2.3.1. Physicochemical Characterization

The physicochemical characterization of DEP and PP01 mucilages included color, betalain content, pH, and reducing sugars content, as well as the methods that will be described in this subsection.

Water activity (Aw): An AquaLab hygrometer (Decagon Devices, Inc, Pullman, WA USA) with a chilled mirror dew point sensor was used.

Soluble (SDF) and insoluble (IDF) dietary fiber: SDF and IDF content in mucilages was determined through the AOAC method 991.43 [48] using a Sigma-Aldrich TDF 100A enzymatic assay kit (St. Louis, MO, USA), SDF and IDF concentrations were corrected by subtracting the residual protein (AOAC method 969.52) and ash (obtained by incinerating samples at 525 °C for 5 h). Results were expressed as grams of dietary fiber per 100 g dry weight (dw).

Proteins were determined with the Bradford reagent using bovine serum albumin as a standard [54].

#### 2.3.2. Structural Characterization

UV–visible spectrophotometric analysis: 5 mg of each mucilage was dissolved in 5 mL of distilled water, followed by centrifugation to clarify the mucilage solution, which was scanned from 200 to 700 nm using a spectrophotometer (Genesys 10 S UV-VIS, Thermo Scientific, Beijing, China).

Fourier transform infrared spectroscopy (FTIR): To characterize and detect the functional groups in the structure of DEP and PP01 mucilages, the spectra in the mid-IR region (4000–500 cm^−1^) of the sample were analyzed with a spectrometer, using a universal attenuated total reflectance (ATR) accessory for powder analysis with the strength indicator set at 70% and an L1600107 single reflection diamond (Spectrum Two FT-IR Perkin Elmer, Bedfordshire, UK) [51,55]. Each sample was analyzed in triplicate at 20 °C. Spectra were analyzed using Spectragryph–optical spectroscopy software, Version 1.2.16.1, 2022, http://www.effemm2.de/spectragryph/ (accessed on 22 October 2024). The area over individual peaks was plotted and integrated over the entire frequency range. A constant straight line at the local maximum point was used as the baseline.

Monosaccharide profile, total carbohydrates, and degree of esterification: The monosaccharides that make up the mucilages were identified by gas chromatography coupled to mass (GC/MS). First, 5 mg of each mucilage, PP01 and DEP, was hydrolyzed with 6.0 N HCl at 85 °C for 24 h. The solution was dried under a stream of air. The trimethylsilylated derivatives of these monosaccharides were formed by reacting them with 100 μL of the reagent N,O-bis (trimethylsilyl) trifluoroacetamide (BSTFA) and 150 μL of pyridine for 1 h at 90 °C. The process to separate the trimethylsilyl derivatives of sugars of hydrolyzed mucilages was carried out on a chromatograph (Perkin Elmer, GC: CLARUS 580 and MS: SQ8S), equipped with an electron impact ionizer and an Elite 5 ms 30 m × 0.32 mm × 0.25 μm capillary column. The chromatographic conditions were as follows: injector temperature 250 °C, N2 flow at 0.55 mL/min; the temperature program was held for 3 min at 140 °C, then raised to 200 °C at 3 °C/min, and finally brought to 300 °C at 15 °C/min. An injection volume of 2 μL was used. The conditions for mass spectrometry were: solvent delay 3 min, MS scan 30 to 500 *m*/*z*, source temperature 250 °C, and transfer line temperature 230 °C. The identification of analytes was based on the coincidence of fragmentation using the NIST libraries integrated into the equipment software. Additionally, in the case of arabinose, rhamnose, galactose, glucose, fucose, and mannose, for which there was an available standard, the coincidence in retention time was recorded. The relative quantification was expressed as the area under the peak of each analyte relative to the sum of the areas of all the peaks, and in molar ratio for each mucilage.

The degree of methyl esterification (DME) was estimated by the ratio of moles of esterified D-galacturonic acid to the sum of the relative moles of unesterified and esterified D-galacturonic acid. The total sugar content was calculated from the sum of all monosaccharide residues determined in the GC/MS analysis, expressed as a percentage [56].

Zeta potential: The zeta potential of the colloidal dispersions was measured using a Zetasizer (Malvern-NanoZS90, Malvern Ltd., Worcestershire, UK). Mucilages were dissolved in distilled water at a concentration of 0.1% (*w*/*v*) with stirring and without pH adjustment. Samples were placed into the folded capillary cell (DTS1070, Malvern) and measured at room temperature. To obtain comparable and representative data, results were recorded as the average of six replicates ± standard deviation [57].

Confocal laser scanning microscopy (CLSM): The presence of betalains, pectin, cellulose, and hemicellulose in DEP and PP01 mucilage powder was evidenced by CLSM (LSM 710 NLO, Carl Zeiss, Oberkochen, Germany). The spectral channel (Spectral Image λ Stack) laser mode 405, 488, 561, and 633 nm [58] was used for autofluorescence detection of betalain molecules, such as betanin [59,60]. Secondary fluorescence images were acquired after staining the mucilages with the exogenous fluorophores 0.01% calcofluor white M2R, fluorescent brightener 28 F3543, Sigma, St. Louis, MO, USA (excitation at 380 nm, emission 475 nm [61]), and 1% ruthenium red, Thermo Scientific Chemicals (excitation at 488 nm, emission 510 nm [62]). Stained samples were washed with absolute ethanol and dried at room temperature. All images were acquired at 10× magnification in RGB color and stored in TIFF format at 512 × 512 pixels. Images were analyzed using the Zen Lite software v. 3.8, Carl Zeiss Microscopy, Oberkochen, Germany.

Environmental scanning electron microscopy (ESEM): ESEM was used to characterize the microstructure of the mucilage extracts. Samples of DEP and PP01 mucilage powder were placed on a double-sided carbon tape adhered to an aluminum sample holder; a colloidal silver contour was set to avoid charge accumulation on the sample. The micrographs were acquired using a SEM-QUANTA FEG-250 scanning electron microscope, Eindhoven, The Netherlands.

### 2.4. Statistical Analysis

Experimental data were expressed as the mean ± standard deviation of at least three replicates. For comparison of means in the different tests and assays, Student’s *t*-tests were used, with the significance level set at 0.05. SigmaPlot software version 14.0.3 was used.

## 3. Results and Discussion

Pulp yield is an important factor for processing prickly pears. After peeling and pulping purple prickly pears, 75.79% of the weight was pulp (including the mesocarp and endocarp), 22.34% was peel (epicarp), and 1.87% was seeds. Variations in the percentage of the peel of prickly pears (*Opuntia* spp.) have been reported, ranging from 30% to 67% of the fruit weight, depending on the variety and cultivation area [3,63,64,65,66]. Therefore, our method of peeling the frozen fruit, which preserved the mesocarp and endocarp for processing, was convenient as it reduces pulp loss from the peel, and, at the same time, it reduces 63% of the fat and 70% of the protein.

### 3.1. Physicochemical Characterization of the MEP

The results of the physicochemical analysis of the MEP are shown in Table 1.

In purple prickly pear MEP, a soluble solids content of 12.17 ± 1.89 °Bx and 14.81 ± 0.54% of total sugars was observed, similar to that reported for the same *O. ficus-indica* cultivar (13.6 °Bx and 18.66 mg glucose/mL) [67] and in three *Opuntia* cultivars (around 14 °Bx) [68]. The total soluble solids content and the low reducing sugars content (1.7%), as well as the relatively high acidity (54 mg citric acid/100 g) are consistent with the ripening stage of the commercial standard (barely pigmented peel). These physicochemical characteristics are similar to those reported by Yahia and Mondragon-Jacobo [69] for 10 Mexican cultivars of *Opuntia* spp. with reported values from 11.6 to 15.30 °Bx.

The pH of the pulp of *O. ficus-indica* cv. San Martín was 5.59, which is consistent with that reported for Mexican cultivars of *O. ficus-indica* such as the orange prickly pear cultivar, the red San Martín cultivar [70], and the white crystalline cultivar [67], which report values close to pH 6, regardless of the ripening state. In the case of *Opuntia streptacantha*, the pH level changes as maturation progresses. This species shows a low pH value (around pH 3) when maturation is incipient, which then increases to pH 6. However, *Opuntia ficus-indica* did not show a change in pH during maturation, with constant values near pH 6 [68]. This observation thus means that prickly pears are a low-acid food.

On the other hand, the high moisture content of the fruit (88.73%) and its low protein and lipid content (0.18 and 0.40%, respectively) are characteristic of this fruit, as is the higher proportion of lipids, protein, crude fiber, and ash in the peel [71].

The MEP had a high content of total sugars (14.81 ± 0.54%), of which 1.73% were reducing sugars. This is explained by the nature of the plant cell wall, which is composed of a complex of polysaccharides such as mucilage, cellulose, hemicellulose, and pectin [72]. Previous work has shown that the lignin content in the skin of the prickly pear is low (2.4% dw) and that its main components are polysaccharides (66.1% dw) such as cellulose (27% dw) [35,71].

Esatbeyoglu et al. in 2015 [60] reported the presence of betalains (betaxanthin and betanin) in the red-purple prickly pear. In the MEP of our samples, 129.7 ± 5.2 mg BE/100 g (dw) and 57.6 ± 3.4 mg IE were recorded. The range of BE and IE values reported by other authors is wide (11.1 to 104.5 BE and 2.93 to 50.1 IE) [73,74]), possibly due to factors such as the cultivar or variety, the stage of maturity, and the climate or geographical location of prickly pear production [75]. However, the BE and IE values reported here are higher than those reported for the *Opuntia ficus-indica* cv. San Martín (50.0 ± 1.3 mg betaxanthins/100 g dw and 104.4 ± 1.7 mg betacyanins/100 g dw) [76].

### 3.2. Separation of Mucilage from the MEP

For the separation of mucilage from the MEP by tangential flow microfiltration and subsequent precipitation (PP01 mucilage), the operating parameters in the membrane system were feed flow rate 4.22 ± 0.03 L/h (equivalent to a mass flow of 3.99 ± 0.26 kg/h) and permeate flow of 1.03 kg/m^2^h, which represents the weight of permeate collected in one hour through the 420 cm^2^ of surface in the membrane, reaching a volumetric reduction factor (VRF) of 1.67 ± 0.17, which is reflected in the turbidity of the fluid, measured in formazin turbidity units. FTU went from 772 FTU in the feed line to 1654 FTU in the retentate. This represents a 2.14-fold increase and indicates that by decreasing the volume of the sample, the concentration of molecules larger than 0.1 µm increased. During the diafiltration stage, the retentate from the separation stage was microfiltered with distilled water to reduce the microsolutes and purify the mucilage by washing. Then, the retentate containing the mucilage was concentrated in the microfiltration module, obtaining a concentrated retentate with a VRF of 2.26. Finally, after precipitating the mucilage in ethanol, drying, and grinding it, a yield of 3.61% dw was obtained. This value is similar to the data reported for mucilage from the peel of *O. ficus-indica* by ethanolic precipitation [34,35], but was achieved with a lower consumption of organic solvents.

In the DEP method, the hydration of the polymer chain segments was interrupted (Guo et al., 2016 [77]), as ethanol competes with polysaccharides for water molecules, causing the precipitation of mucilages. With our methodology, a yield of 9.92% dw was reached, which was higher than the yield reported for mucilage from the peel of red-orange fruits of *O. ficus-indica* [34,35] and that reported for the mucilage from *Opuntia stricta* Haw peel [78].

### 3.3. Physicochemical Characterization of DEP and PP01 Mucilages

Table 2 shows the physicochemical characterization of the mucilages of *O. ficus-indica* cv. San Martín, obtained by direct ethanolic precipitation (DEP) and by tangential microfiltration (PP01). PP01 mucilage contains 15 % more moisture than DEP mucilage; however, it has a 34 % higher water activity, which may be the effect of its lower total sugars content (42.6 % less than DEP mucilage). This may be because polysaccharides (formed by monomeric sugar units) are generally hydrophilic solutes capable of interacting with water molecules, retaining them, and decreasing their water vapor pressure and water activity [79]. As a result, the amount of free water available for chemical reactions and microbial growth is reduced [80].

The PP01 mucilage presented a hue of 66.30° within the chromatic circle (closer to yellow than to red) and chromaticity of 10.48 (i.e., just a touch of color) (scale 0 to 100); these values reflect the low content of betacyanins and betaxanthins, as shown in Table 2, and represent the loss of 98.4% of the betacyanin pigments and 96.5% of betaxanthin from the MEP. This loss is attributable to the diafiltration stage of the separation and purification process of the mucilage, and to ethanol precipitation which solubilizes betalains. On the other hand, the remarkable color difference [47] between the DEP and PP01 mucilages (ΔE of 26.82) highlights the retention of 51.9% betacyanin in the DEP mucilage which, together with the concentration at 125% of betaxanthin (reflected by the hue and chromaticity values, 280.13° and 26.46, respectively) give the DEP mucilage a bright purple color that could be attributed to an entrapment of the pigments during the precipitation in ethanol.

As shown in Table 2, the composition of PP01 and DEP mucilages was dominated by total sugars (66.43% and 94.71%, respectively). The PP01 mucilage contained 18% more insoluble fiber (probably lignin, cellulose, and hemicellulose type B) than the DEP mucilage. However, in the DEP mucilage there was 20.5% more soluble fiber (probably hemicellulose type A, pectins, and mucilages) than in the PP01 mucilage. This represents a more balanced ratio of SDF/IDF in the DEP mucilage (1:1.3) than in the PP01 mucilage (1:2.0), which could contribute to improving the functionality of dietary fiber and provide significant health benefits [81].

The PP01 mucilage had 4.5 times fewer proteins (0.263 ± 0.018%) than the DEP mucilage (1.173 ± 0.088%), which is attributed to the diafiltration stage of the PP01 mucilage, which involves the separation of small molecules (monosaccharides, salts, and peptides) through the membranes, while molecules are put in circulation with fresh solvent, similar to membrane dialysis. Similarly to our findings, Salehi et al. in 2019 [82] reported 0.86 ± 0.03% protein content in mucilage of the fruit of *O. ficus-indica* obtained by microwave-assisted extraction, which could have favored the separation of proteins from the mucilage during ethanolic precipitation. On the other hand, using the Bradford method, in the mucilage obtained from the skin and fruit of *O. ficus-indica*, we did not detect proteins [5,34]. The high ash content in the PP01 mucilage (23.3%) may be due to the concentration of minerals by entrapment in the mucilage during tangential microfiltration and the subsequent co-precipitation with the mucilage during alcoholic precipitation. This is consistent with that reported for the mucilage of *O. ficus-indica* cladodes obtained by ultrafiltration [27]. The lower mineral content in the DEP mucilage (0.5%) is consistent with the low mineral content in citrus pectin obtained after repeated washing of the precipitate with 96% alcohol [21,83].

### 3.4. Structural Characterization

#### 3.4.1. UV–Visible Spectrophotometric Analysis

Figure 1 shows the absorption spectra of the PP01 and DEP mucilages presented as an area plot, together with the first numerical derivative of the absorption spectrum as a line graph to show the wavelengths with the maximum absorbances. By comparing these absorption spectra with those of chromophores reported in the literature, a correlation between these values and the presence of particular molecules can be established.

Figure 1 shows an intense peak with maximum adsorption at 202 nm in the UV–vis spectra of the mucilages PP01 and DEP. This peak represents the major component attributed to carbonyl groups (C=O) present in aldehydes, uronic acids, or esters, which are characteristic of polysaccharides such as cellulose, pectins, or cell wall hemicelluloses in vegetables [84,85]. The above indicates the presence of a common functional group base and denotes their polysaccharide nature. The absorption peak at 234 nm is attributable to phenolic compounds [86]. In the ultraviolet region, a maximum absorbance at 286 nm was also evident due to the cyclo-DOPA residue in the betacyanin molecule [75]. Finally, a small peak with maximum at 532 nm is related to pigments of the betalain group (max: 520–550 nm), betacyanins that are characteristic of the purple color of *Opuntia ficus-indica* fruits [25,87]. The absorbance of this small peak is 3.7 times higher in the DEP mucilage (0.1817) than in the PP01 mucilage (0.049), which is consistent with our analysis of the chemical composition and color parameters.

#### 3.4.2. Fourier Transform Infrared Spectroscopy (ATR–FTIR)

In the FTIR spectra of the DEP and PP01 mucilages, the characteristic functional groups of the major components in plant cells such as cellulose, hemicellulose, lignins, pectins, and betalain-type pigments were traced [34,78,88]. The FTIR spectra of both extracts are shown in Figure 2.

In both FTIR spectra (Figure 2), a big and wide absorption band is observed at approximately 3270 cm^−1^. This band is attributed to the stretching vibrations of hydrogen bridges between water and -OH groups in alcohol and carboxylic acid, which are generally involved with intermolecular hydrogen bridges on polysaccharides [34,89]. This way, the greater area over the curve in DEP mucilage relative to PP01 mucilage indicates a higher number of -OH groups engaged in interactions with the moisture of the polysaccharide (10.75 % in PP01 mucilage and 9.16% in DEP). This could also explain the lower water activity of DEP mucilage (Aw 0.27) relative to PP01 mucilage (Aw 0.41). Consequently, the greater proportion of polysaccharides in DEP mucilage reduces the water vapor pressure and the availability of water to participate in reactions. The band at 2916 cm^−1^ is also present in both mucilages produced by CH stretching vibrations of methine groups generally present in hexoses (e.g., glucose or galactose) or deoxyhexoses (e.g., rhamnose or fucose) [37,90].

The carbonyl bond stretching bands around 1730 cm^−1^ and 1600 cm^−1^ are characteristic of uronic acids, with esterified and non-esterified carboxyl groups, respectively, as reported in pectins [91,92,93]. Table 3 shows a comparison of the area over the curve of these bands. In both DEP and PP01 mucilages, 88% of the carbonyl groups are free and only 12% are esterified. In the DEP mucilage, the area over the curve is approximately 35% greater than in the PP01 mucilage for these functional groups. In addition, both FTIR spectra show a set of bands around 1380, 1316, and 1240 cm^−1^, which correspond to the symmetric bending of the methyl group, the stretching of the CO bond of the carboxyl group present in uronic acids, and the stretching of the CH bond, respectively. The presence of these bands is in keeping with the report for hemicellulose obtained by alkaline ethanol extraction of eucalyptus wood [94]. These findings are consistent with the higher concentration (20.5%) of soluble fiber in the DEP mucilage relative to the PP01 mucilage. In this case, soluble fiber would correspond to type A hemicellulose, pectins, or mucilages, as reported [94,95]. Finally, the area above the curve of the band at 890 cm^−1^, associated with β bonds in the polysaccharide structure [82], shows no significant difference (p < 0.05) between both mucilages, suggesting a similar proportion of structural polysaccharides, which coincides with the amount of total fiber 61.44 and 59.28% dw for the PP01 and DEP mucilages, respectively (Table 2).

The stretching band of the C–O–C bond (aryl–alkyl ether bond) at 1240 cm^−1^ has lower transmittance in the DEP mucilage than in the PP01 mucilage; this was attributed to the union between the betalamic acid bound to cyclo-dopa 5-O-glucoside in the betanin structure (red betalains), which is consistent with the hue (H°) of 280.13 ± 0.58 corresponding to the red-purple color in the DEP mucilage (Table 2).

It is possible that some of the absorption bands mentioned above overlap with bands associated with functional groups of betalains; for example, the absorption band at 1600 cm^−1^, attributable to C=C double bonds of the chromophore group in the betalain molecule; the NH stretching bands around 890 cm^−1^; and the 3270 cm^−1^ band and the one at 1015 cm^−1^ attributable to the stretching of the CN bond, both of which are associated with the amine in betalamic acid, as reported for red beet and *Bougainvillea glabra* extracts [88,96]

#### 3.4.3. Total Carbohydrates, Monosaccharide Profile, and Degree of Esterification

Through an exploratory analysis using HPLC, the reference standards were compared with the residues of hydrolyzed monosaccharides. In the PP01 mucilage, the predominant monomers were glucuronic (retention time 6.784 min and 54.98% of area) and galacturonic acids (retention time 7.372 and 37.24% of area). In the DEP mucilage, the predominant monomer was galacturonic acid (retention time 6.760 and 60.48% of area). In turn, the GC/MS (IE) analysis of the trimethylsilylated derivatives obtained from the PP01 and the DEP mucilages showed the presence of eight major neutral monosaccharides: arabinose, rhamnose, xylose, fucose, mannose, galactose, glucose, and myo-inositol, as well as galacturonic and glucuronic acids, and the galacturonic acid ester. Table 4 lists the monosaccharide–TMS residues identified by GC/MS, retention time (RT), relative area of each peak, molar ratio, and coincidence factor.

Qualitatively, the polysaccharides in PP01 and DEP present a monosaccharide profile consistent with mucilages [97], containing L-arabinose, D-galactose (pyranose and furanose forms), D-xylose, L-rhamnose, glucuronic acid and, as the main monomer, D-galacturonic acid (Table 4); however, there are differences in the molar ratio. This is clear in galacturonic acid, with 14.95 for PP01 mucilage and 33.65 for DEP [97]; however, it does not coincide with the pectin-type polymers previously reported in polysaccharides extracted from prickly pear peel [5,40]. Additionally, the trimethylsilylated derivative of D-galactose without anomeric –OH at carbon 1, (RT 13.718 min and 13.803 min in the PP01 and the DEP mucilages, respectively) suggests it originates from a non-reducing terminal residue or a branching point, indicating the existence of non-linear polymeric structures such as mucilage [97]. It is worth noting that the DEP mucilage showed twice as much galacturonic acid as the PP01 mucilage (molar ratios of 33.65 and 14.95, respectively). On the other hand, Salehi et al. in 2019 [82] used microwave-assisted aqueous extraction to obtain a polysaccharide fraction with arabinoglucan structure from *O. ficus-indica* fruit that was mainly composed of glucose (78.0%), arabinose (12.2%), xylose (4.8%), galactose (2.4%), and mannose (2.4%). This composition is consistent with the sugar residues in the PP01 mucilage but not in the DEP mucilage, where the amount of mannose was not quantifiable, and the proportion of glucose was 25% lower than that in the PP01 mucilage. From these observations, we deduce that this type of arabinoglucan was retained by the 0.1 µm microfiltration membranes and was partially removed during DEP. By summing the monosaccharide–TMS residues of the PP01 and DEP mucilages, the total carbohydrate content was estimated at 66.43% and 94.71%, respectively. The total carbohydrate was proportionally lower in the PP01 mucilage due to its high mineral content (23.32%) compared to the DEP mucilage (0.51%).

The DME estimated from the ratio between the area under the curve of the peaks was assigned to esterified and free galacturonic acid; in the GC/MS mass spectrometries of the PP01 and DEP mucilages, the area under the curve was 15.65% and 22.01%, respectively. There was 40% more esterified galacturonic acid in the DEP mucilage. In the PP01 and DEP mucilages, the degree of esterification was considered low for a pectin-type mucilage, indicating the presence of a significant number of acid groups (-COOH). This feature furnishes these polymers with a polar and hydrophilic character, and the possibility of interacting with divalent ions through the carboxyl acid groups that could form gels stabilized by an egg-box mechanism. Low esterified pectins have also been reported in beet, sunflower, and purple prickly pear (DME < 50%), unlike the pectins of apple, citrus, and green and orange cultivars of *Opuntia ficus-indica* which are highly esterified (DME > 50%) [21,38].

#### 3.4.4. Zeta Potential

The zeta potential was investigated to estimate the electrical charge on the surface of the PP01 and DEP mucilages, as an indicator of the stability of colloidal dispersions (Table 5.

The zeta potential of the PP01 mucilage dispersion, which ranged from –30.47 to –34.93 Mv, was not statistically different (*p* = 0.134) from the zeta potential of the DEP mucilage, which ranged from –33.04 to –35.9 Mv, reflecting both its anionic nature in distilled water [57], which is a consequence of the high content of uronic acids (PP01 mucilage 31.31% and DEP mucilage 47.48%) and the low degree of esterification (PP01 mucilage 15.65% and DEP mucilage 22.01%). Generally, in fruit pectins, such as apple, citrus, and sugar beet mucilages [98], half of the GalA residues are esterified, which would reduce the electrostatic repulsion between molecules, giving rise to zeta potentials lower than those presented by the PP01 and DEP mucilages. Thus, a low degree of esterification and a high content of dissociated uronic acids contributes to the electrostatic repulsion between the mucilage molecules, increasing the stability of the dispersions with a zeta potential >30. Despite the complexity of the composition of the PP01 and DEP mucilages (different sugar residues in polysaccharides, minerals, betalains, and water content), the stability of the dispersions is not affected, as was reported by other authors [99].

#### 3.4.5. CLSM

Fluorescence imaging in CLSM was used to monitor common polysaccharides in the plant cell wall of *Opuntia* sp. fruits, such as cellulose, hemicellulose, lignin, pectin, and betalains [59,100,101]. We used the lambda function of the confocal microscope, which consisted of scanning with four wavelengths (633, 561, 488, and 405 nm) to identify the spectral channels that emit fluorescence without the need for external fluorochromes. Figure 3A corresponds to the autoflorescence of the PP01 mucilage emitted with maximum intensity in the green region of the electromagnetic spectrum at 501 nm, which, according to the literature, is related to betaxanthin (yellow betalain) [102,103,104]. In the DEP mucilage (Figure 4B), this fluorescence is also present due to betaxanthin (shown in green pseudocolor). Another spectral channel was observed in the yellow region with a maximum at 597 nm; this is attributable to betanin (red pseudocolor), which is considered the most abundant betacyanin in red prickly pears [60]. These results are in keeping with the quantification of betalains shown in Table 2 and our analysis of the UV–vis and FTIR spectra.

Figure 4A,B show micrographs of the PP01 and DEP mucilages obtained using exogenous fluorophores (calcofluor white, CFW) with specific binding affinity for β-glucans with β-1,3 and β-1,4 bonds, revealing their presence on a large surface of both samples. This type of bond is present in polysaccharides such as cellulose, xyloglucans, and carboxylated polysaccharides [105,106,107]. Finally, Figure 4C,D show micrographs of the PP01 and DEP mucilages, respectively, stained with ruthenium red (RR). Practically the entire sample emits fluorescence, revealing the presence of uronic acids in the mucilages. The RR is specific for pectic substances, mucilages, and gums, as it produces a stereoselective staining for uronic acids due to a positive charge. For instance, in pectins, RR can be fixed to the surface of rhamnogalacturonan by interacting with its negative charge [62]. These CLSM images are consistent with the high concentration of uronic acids shown in Table 4 and with the presence of organic acids evidenced by the analysis of the UV–vis and FTIR spectra. Moreover, these results show that the uronic acids are homogeneously distributed in both mucilages.

#### 3.4.6. ESEM

The morphology of the polysaccharides present in the PP01 and DEP mucilages was examined using a scanning electron microscope (SEM). The PP01 mucilage in Figure 5A showed a surface of distributed flaky aggregates with an irregular shape and size and smooth surfaces, similar to the structures reported in the mucilage of the peel of *O. ficus-indica* fruit [108] and in the mucilage of the peel of *Opuntia dillenii* haw, obtained by microwave-assisted extraction [37]. It has been proposed that the differences in microstructure in this type of mucilage can be attributed to the extraction method.

In Figure 5B, the DEP mucilage presented an even rougher, more porous and cracked surface than the PP01, with aggregates of irregular shape and size.

## 4. Conclusions

The processing of the mesocarp–endocarp assembly of *Opuntia ficus-indica* cv. San Martín (76% of the fruit) reduced the loss from the rind while removing 63% of the fat and 70% of the protein, without the use of organic solvents. The epicarp of the purple prickly pear contains the highest proportion of lipids, protein, ash, and crude fiber in relation to the fruit as a whole. These by-products could be used, for example, as a source of dietary fiber or as a food colorant.

The mesocarp–endocarp pulp (MEP) showed a high content of purple-red betalains, 129.7 mg betanin equivalents/100 g dw, and orange-yellow betalains, 57.6 mg indicaxanthin equivalents/100 g dw. Of these betalains, 97% were separated from the mucilage by tangential microfiltration (PP01) and can be recovered in the filtration line for potential use as a functional ingredient.

Two mucilages were obtained, one by tangential microfiltration (PP01) and another by direct ethanol precipitation (DEP), with significant differences (*p* < 0.05) in hue (H°), chromaticity, betacyanin content, betaxanthin, total sugars, and proteins. In all these cases, the lowest values corresponded to the PP01 mucilage, reflecting purification during microfiltration. On the other hand, the PP01 mucilage had an ash content 46 times higher than the DEP mucilage. This reflects the retention of minerals in the form of calcium oxalate crystals in the polysulfone membrane module. Similarly, no significant differences in insoluble (41.21%) and soluble (20.23%) dietary fiber content were found between the mucilages in either process.

The yield of mucilage obtained by tangential microfiltration (3.61%) was similar to that reported in the literature for *O. ficus-indica* peel pectins and mucilages obtained by aqueous extraction followed by precipitation in ethanol. On the other hand, the yield of mucilage obtained by direct precipitation in ethanol (9.92% dw) was higher, reflecting the effect of the inclusion of the mesocarp in the alcoholic extraction.

FTIR spectra showed numerous hydrogen bonds between water and -OH groups in alcohol and carboxylic acids in DEP mucilage, which is related to its lower water activity (Aw 0.27) compared to the PP01 mucilage (Aw 0.41). A degree of esterification of less than 50% was also found in the PP01 and DEP mucilages, which makes them like low methoxyl pectins.

The monosaccharide profile determined by GC/MS analysis in the PP01 and DEP mucilages is composed of L-arabinose, D-galactose (in pyranose and furanose forms), D-xylose, L-rhamnose, D-fucose, beta-D-glucopyranose, with a high content of uronic acids (31.3% in PP01 and 47.5% in DEP) and a low degree of esterification (15.7% in PP01 and 22.0% in DEP), which explains its anionic nature in distilled water (zeta potential in PP01 = 39.7 MV and in DEP = 34.47 MV) and opens the possibility of forming gels stabilized with divalent ions through the carboxylic acid groups by an egg-box mechanism, with applications in the food and pharmaceutical industries.

## Figures and Tables

**Figure 1 polymers-16-03383-f001:**
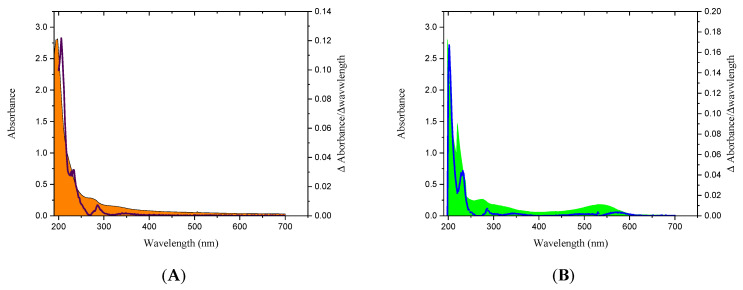
UV–visible absorption spectra from 200 to 700 nm of purple *O. ficus-indica* mucilages extracted by microfiltration–precipitation (**A**) (PP01 mucilage), and direct ethanolic precipitation (**B**) (DEP mucilage).

**Figure 2 polymers-16-03383-f002:**
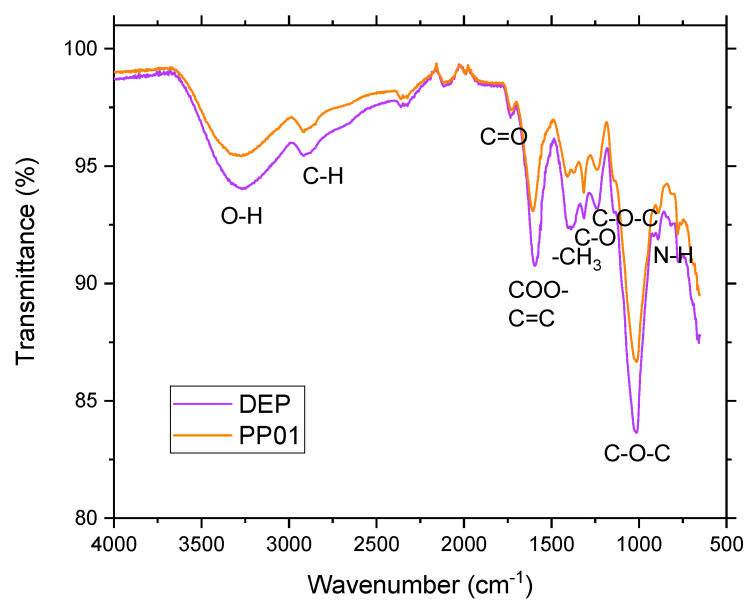
FTIR spectra of mucilages extracted by direct ethanol precipitation (DEP) and by microfiltration (PP01) from *Opuntia ficus-indica* cv. San Martín.

**Figure 3 polymers-16-03383-f003:**
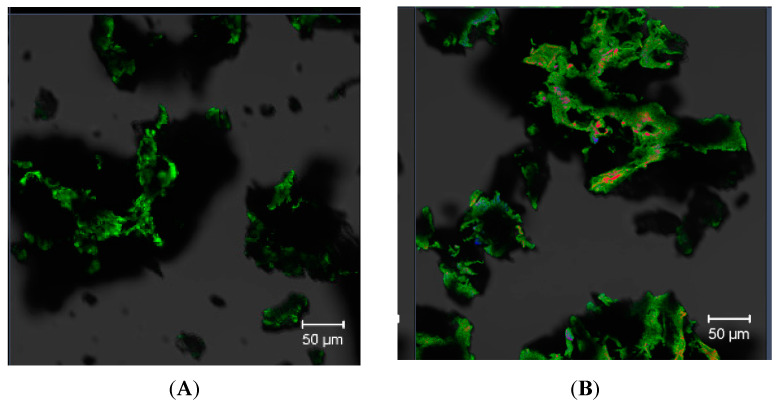
CLSM images of betalain autofluorescence in the PP01 mucilage (**A**) and DEP mucilage (**B**). Scale bar = 50 µm.

**Figure 4 polymers-16-03383-f004:**
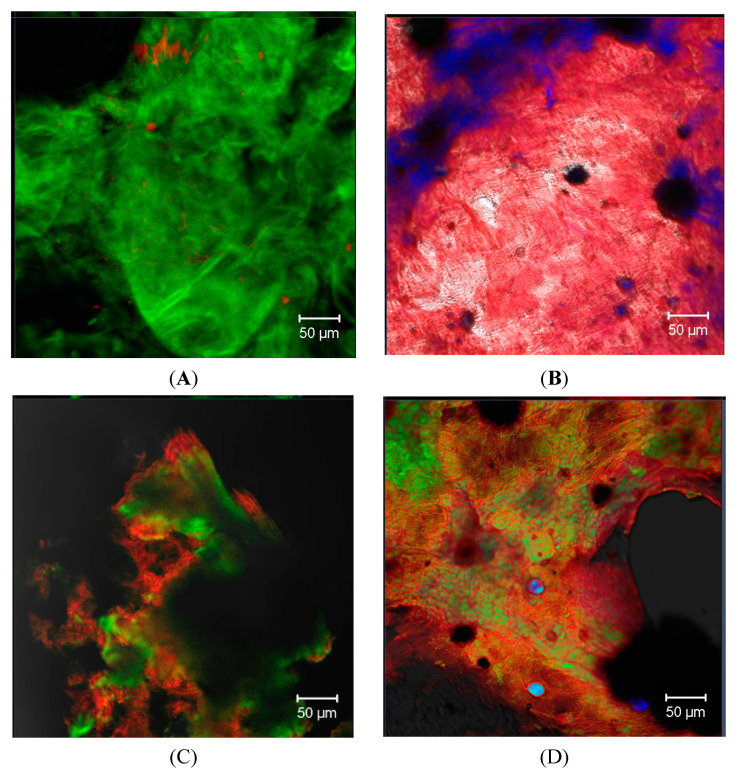
CLSM images of *Opuntia ficus-indica* fruit mucilages obtained by tangential microfiltration (**A**) and ethanol precipitation (**B**), stained with CFW, showing the presence of β-glucans. Mucilages were stained with ruthenium red to reveal the presence of polysaccharides with uronic acids; (**C**) PP01 and (**D**) DEP mucilage. Scale bar = 50 µm.

**Figure 5 polymers-16-03383-f005:**
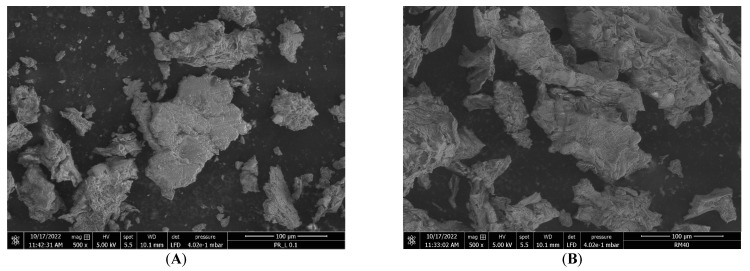
SEM images of purple *Opuntia ficus-indica* mucilages obtained by (**A**) tangential microfiltration (PP01 mucilage), showed flaky aggregates and (**B**) direct ethanol precipitation (DEP mucilage) showed porous and cracked surface. Scale bar = 100 µm.

**Table 1 polymers-16-03383-t001:** Physicochemical analysis of the peel and pulp of purple prickly pear (*Opuntia ficus-indica* cv. San Martín).

Feature	Peel	Pulp
Moisture content (% *w*/*w*)	83.07 ± 0.06	88.73 ± 0.06
Fat (% *w*/*w*)	0.70 ± 0.01	0.40 ± 0.06
Protein (N × 6.25) (% *w*/*w*)	0.41 ± 0.03	0.18 ± 0.02
Ash (% *w*/*w*)	1.37 ± 0.02	0.36 ± 0.01
Crude fiber (% *w*/*w*)	1.48 ± 0.04	0.15 ± 0.02
Nitrogen-free extract (% *w*/*w*)	12.97 ± 0.1	10.18 ± 0.05
Acidity (mg citric acid/100 g)	Nd	54 ± 12
Total soluble solids (°Bx)	Nd	12.17 ± 1.89
pH		5.59
Total sugars (% *w*/*w*)	Nd	14.81 ± 0.54
Reducing sugars (% *w*/*w*)	Nd	1.73 ± 0.064
Betacyanins (mg BE/100 g (dw))	Nd	129.7 ± 5.23
Betaxanthins (mg IE/100 g (dw))	Nd	57.6 ± 3.37
L*	27.14 ± 4.79	9.09 ± 0.62
Chroma (C)	16.38 ± 2.49	36.00 ± 0.38
Hue (°H)	50.48 ± 11.20	15.95 ± 0.59

The mean of three replicates ± standard deviation is shown. Color space parameters: lightness (darkness–lightness: 0–100), chroma (saturation: 0–100), and hue angle (0° red, 90° yellow, 180° green, 270° blue). BE, expressed as betanin equivalents; IE, expressed as indicaxanthin equivalents; Nd, not determined.

**Table 2 polymers-16-03383-t002:** Physicochemical analysis of purple prickly pear mucilage (*Opuntia ficus-indica* cv. San Martín) obtained by tangential microfiltration (PP01) or direct ethanolic precipitation (DEP).

Feature	PP01	DEP
Moisture content * (%)	10.746	9.168
Aw (25 °C)	0.41 ± 0.04 ^a^	0.27 ± 0.03 ^b^
L*	40.28 ± 2.0 ^a^	34.19 ± 1.19 ^b^
a*	4.16 ± 0.07 ^a^	26.05 ± 0.53 ^b^
b*	9.60 ± 1.54 ^a^	−4.66 ± 0.37 ^b^
Chroma	10.48 ± 1.45 ^a^	26.46 ± 0.59 ^b^
Hue (H°)	66.30 ± 2.91 ^a^	280.13 ± 0.58 ^b^
∆E	26.82	
Betacyanin (mg BE/100 g dw)	2.122 ± 0.071 ^a^	67.320 ± 15.451 ^b^
Betaxanthin (mg IE/100 g dw)	2.045 ± 0.068 ^a^	72.018 ± 0.043 ^b^
Insoluble dietary fiber (IDF) (% dw)	41.21 ± 1.06 ^a^	39.20 ± 11.74 ^a^
Soluble dietary fiber (SDF) (% dw)	20.23 ± 6.01 ^a^	22.80 ± 2.70 ^a^
Total sugars (%)	66.43	94.71
Reducing sugars (%)	Nd	0.006 ± 0.001
Protein (%)	0.263 ± 0.018 ^a^	1.173 ± 0.088 ^b^
Ash * (%)	23.32	0.51

Color space parameters: Lightness (darkness–lightness: 0–100), chroma (saturation: 0–100), and hue angle (0° red, 90° yellow, 180° green, 270° blue). BE, expressed as betanin equivalents; IE, expressed as indicaxanthin equivalents. * Determined by TGA analysis. Nd. Not determined. In each row, means followed by different superscript letters are significantly different (*p* < 0.05).

**Table 3 polymers-16-03383-t003:** Functional groups and area over the absorption band curve in FTIR spectra of the DEP and PP01 mucilages of *O. ficus-indica* cv. San Martín.

DEP		PP01		
Wavenumber (cm^−1^)	Area over the Curve	Wavenumber (cm^−1^)	Area over the Curve	Functional Group
3267	−422 ± 166 ^a^	3297	−67 ± 31 ^b^	N–H stretching vibrations in secondary amines and H–O–H stretching vibrations
3218	−1136 ± 363 ^a^	3275	−110 ± 35 ^b^	O–H stretching vibrations
2916	−1161 ± 312 ^a^	2919	−428 ± 46 ^b^	asymmetric stretching vibrations in –CH2-
2803	−297 ± 92 ^a^	2838	−260 ± 134 ^a^	symmetric stretching vibrations in –CH2-
1734	−146 ± 14 ^a^	1735	−97 ± 2 ^b^	C=O stretching vibration of carbonyl ester
1596	−1173 ± 104 ^a^	1607	−712 ± 230 ^b^	asymmetric C=O stretching vibration of the free carboxyl group (COO-)
1404	−507 ± 45 ^a^	1410	−412 ± 93 ^a^	symmetrical bending vibration –CH2-CO-
1376	−309 ± 00 ^a^	1375	−192 ± 63 ^b^	vibration of the O-CO-CH3 bond
1314	−322 ± 27 ^a^	1316	−311 ± 101 ^a^	stretching vibration of the CO bond of the carboxyl group
1240	−566 ± 45 ^a^	1239	−357 ± 106 ^b^	C_aromatic_ stretching vibration in -O, C–O–C, aryl–alkyl ether bond
1014	−2449 ± 175 ^a^	1014	−2269 ± 711 ^a^	COC stretching vibration in pyranose, CN stretching
892	−335 ± 24 ^a^	890	−380 ± 116 ^a^	C1–H bending vibrations in β-glycosidic bond
817	−379 ± 4 ^a^	818	−153 ± 33 ^b^	bending vibrations of –NH- bonds in primary and secondary amines
776	−413 ± 160 ^a^	779	−434 ± 262 ^a^	aromatic hydrogen stretching vibration

Values represent the mean of three replicates ± standard deviation. Within rows, means ± standard deviation followed by different superscript letters are significantly different (*p* < 0.05).

**Table 4 polymers-16-03383-t004:** Composition of monosaccharides analyzed by GC/MS in mucilage of purple *O. ficus-indica* cv. San Juan obtained by tangential microfiltration (PP01) and by ethanol precipitation (DEP).

	PP01				DEP			
Probable Molecule	RT (min)	Area % **	Molar Ratio	R	RT (min)	Area % **	Molar ratio	R
beta-L-Arabinopyranose	7.02	0.998	1.36	834 *	7.09	1.152	1.62	836 *
alpha-L-Rhamnopyranose	7.3	2.265	3.37	967 *	7.4	2.908	4.48	950 *
beta-L-Xylopyranose	7.75	1.116	1.52	938	7.85	1.478	2.08	971
D-Fucose	8.486	0.672	1.00	*	8.596	0.649	1.00	*
beta-L-Rhamnopyranose	8.891	2.767	3.77	949	9.001	4.287	6.04	966
Manose	12.147	0.673	1.10	*	ND	ND	ND	ND
beta-D-Galactofuranose	12.722	1.872	3.06	897	12.807	5.991	10.13	916
alpha-D-Xylofuranose	13.418	0.86	1.17	837	13.518	1.060	1.49	983
D-Galactose	13.718	3.725	6.08	973 *	13.803	6.090	10.30	977 *
beta-D-Glucopyranose	14.663	6.830	11.15	972 *	14.748	4.907	8.30	958 *
D-Galacturonic acid	15.018	7.167	12.61	978	15.123	14.399	26.24	978
D-Galacturonic acid methyl ester	15.173	1.33	2.34	918	15.263	4.064	7.41	
beta-D-Glucuronic acid	17.664	9.292	16.35	975	17,739	7.595	13.84	976
Myo-inositol	20.371	0.946	1.54	941	20.446	1.056	1.79	954
Total sugar			66.43				94.71	
Acid sugars			31.31				47.48	
Degree of esterification			15.65				22.01	

* Verified against a standard. ND Not detected. ** Results are expressed as relative percentile of total peak area.

**Table 5 polymers-16-03383-t005:** Zeta potential of purple *O. ficus-indica* mucilages extracted by tangential microfiltration (PP01) and direct ethanolic precipitation (DEP).

Parameter	PP01	DEP
ZP (Mv)	−32.7 ^a^ ± 2.23	−34.47 ^a^ ± 1.43
Mob µm cm/Vs	−2.561 ^a^ ± 0.17	−2.700 ^a^ ± 0.11

Mean of 6 replicates ± standard deviation. Within rows, means followed by different letters are significantly different (*p* < 0.05).

## Data Availability

The original contributions presented in the study are included in the article. Further inquiries can be directed to the corresponding authors.

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
