# Peer review of "Extraction of Purple Prickly Pear (Opuntia ficus-indica) Mucilage by Microfiltration, Composition, and Physicochemical Characteristics"

_polymers, 2024, doi:10.3390/polym16233383_

Round 1

Reviewer 1 Report

Comments and Suggestions for Authors

The paper (Extraction of purple prickly pear mucilage by microfiltration, composition, and physicochemical characteristics) is well written and the study is considered novel. This manuscript investigates the effect of cross-flow microfiltration on the properties of ethanol precipitated mucilage. The experimental design and analyses were found to be compatible and appropriate to the topic of the article. The manuscript needs improvement in the suggested following  issues.

-Please include in the introduction gum or hydrocolloid or mucilage seperation studies carried out in the literature with this process.

-Please detail the purpose of the targeted process in the separation process.

-Line 243 SigmaPlot?

-Line 321: Please comment on the increase in humidity and water activity.

- A significant increase in the amount of ash will adversely affect the functional properties of mucilage? Isn't this a new problem in a study where an economic benefit is desired?

-Line 375: Can you comment on the characteristic carbohydrate peak and how did your method affect the carbohydrate content?

Line 395: Discuss the peak at 3270 in relation to the change in water content.

- It is considered that rheological analyses are necessary to interpret the differences in the functional properties of mucilage.

- In the conclusion, please elaborate on the functional benefits of the method applied to the mucilage, if the purpose of this method is not purification, where the yield is significantly reduced.

Comments on the Quality of English Language

It is sufficient.

Author Response

Point-by-point response to reviewer's comments uploaded as Word file.

Reviewer 2 Report

Comments and Suggestions for Authors

The work presented is well structured and rich, but in the abstract, in the introduction and especially in the conclusions there is a lack of interpretation in perspective of previous studies and of the working hypotheses.

·         In the title and in the abstract, as well as in the text, it would be good to write not only the English and common name of the prickly pear cactus, but also the scientific name, Opuntia ficus indica. The reason  for this is that it will be easier to find this article if someone is looking for information about this mucilage. ( most of your reported refences have the scientific name in the title).

·         Throughout the article, the semicolon is often used where it would be sufficient and more correct to use only the comma.

·         In the abstract and especially in the introduction, the purpose of the paper and its meaning is not well defined.

·         In line 36 and in the entire text, refences are given as [2], [3], [4]. This is not correct. The right way to report them is [2-4].

·         In Materials and methods, it is not clear which part of Opuntia ficus indica you used. At the end of paragraph, I understood that it was probably just the fruit, but in the line you reported „The prickly pears were weighed, washed, and sanitized….“, as if you weighed, washed, and sanitized the whole plant. Please, indicate which parts you actually used and tested

·         In lines 103-104 you have indicated the values of the hue angle (H°) as 0º to 360º. They are expressed in degrees so the correct form has to be 0°to 360°. Can you better explain this sentence “To bring Hº to the interval ranging from 0º to 360º, the constant 360 is summed if the result is negative“?

·         Line 125: What methods you modified? Perhaps you mean that the MEP polysaccharides were separated by ethanol precipitation, modifying some of the methods already known in literature.

·         Line 139: write the mass in grams better as a number

·         Lines 168-169: Water activity (WA). An AquaLab hygrometer (Decagon Devices, Inc, USA) with a 168 chilled mirror dew point sensor was used. This is what you reported. It would be better to write Water activity (WA): An AquaLab hygrometer (Decagon Devices, Inc, USA) with a 168 chilled mirror dew point sensor was used. or To estimate water activity (WA), an AquaLab hygrometer (Decagon Devices, Inc, USA) with a 168 chilled mirror dew point sensor was used.

The same applies to the rest of the paragraph.

·         Table 1: the title of the left column cannot be “component” because you reported also, pH, Chroma,…. Maybe better “physico-chemical property” or “feature” as indicated in Table 2?

In the same table some of the symbols used need to be checked again, i.e. not everywhere you have written ±.  The table has a footer, but it is not indicated in the table.

·         Table 3: you have given some superscript a, b, but no explanation. Titles of each columns could be written in smaller font

·         Table 4: the same comment as for table 3

·         Conclusions: too short, it should be implemented. The differences between the mucilages obtained by the two different methods are reported but the comments on these evidences are missing.

·         Author Contribution: It is not necessary to write the full surname of the authors , only the initials such as X.X. and Y.Y…

·         References: they are not reported as required in the templates. For example: Journal Articles:

1. Author 1, A.B.; Author 2, C.D. Title of the article. Abbreviated Journal Name Year, Volume, page range.

Author Response

(The authors gave the same response as above.)
